# Exploring How the Psychological Resilience of Residents of Tourism Destinations Affected Brand Ambassador Behavior during the COVID-19 Pandemic

**DOI:** 10.3390/bs12090337

**Published:** 2022-09-15

**Authors:** Haihong Wang, Litong Liu, Hongxia Sha

**Affiliations:** Department of Tourism Management, School of Business, Liaoning University, Shenyang 110036, China

**Keywords:** psychological resilience, brand ambassador behavioral intentions, resident–tourist interaction, cognitive reappraisal

## Abstract

In order to focus on the degree of adaptation and resilience of residents of tourist destinations to the changes caused by the COVID-19 pandemic, and to explore how the psychological resilience of residents affects their attitudes and behavioral intentions toward destination brand development, this study constructs a structural equation model guided by positive organizational behavior and uses a questionnaire method to conduct the research. The results show that (1) residents’ psychological resilience has a positive effect on brand ambassador behavioral intentions; (2) residents’ psychological resilience has a positive effect on resident–tourist interaction; (3) resident–tourist interaction has a positive effect on brand ambassador behavioral intentions; (4) resident–tourist interaction plays a mediating role between psychological resilience and brand ambassador behavioral intentions; and (5) cognitive reappraisal plays a moderating role between psychological resilience and resident–tourist interaction. The findings not only fill the deficiency of positive organizational behavior in tourism research, but also provide a theoretical basis for developing residents as destination brand ambassadors in the context of the COVID-19 pandemic according to destination branding. In fact, destination managers not only need to strengthen residents’ behaviors to participate in destination brand development, but also should care about the psychological state and emotional events of residents who are negatively affected by the COVID-19 pandemic.

## 1. Introduction

Since the outbreak of the COVID-19 pandemic, the tourism industry has been in a state of “stop and go,” and many tourism-related businesses and employees have suffered significant losses. During the relatively stable phase of the epidemic, scenic spots were opened one after another, and as many residents of the destination were involved in tourism operations, they inevitably had to come into contact with tourists from different regions. This greatly increases the risk of contracting the COVID-19 virus [1]. This double pressure from mental and financial sources will cause serious psychological damage to destination residents, making it difficult for them to devote themselves to their work [2,3]. Therefore, the psychological adaptation and resilience of destination residents becomes critical and could well avoid some of the negative effects.

As the concept of tourism and leisure takes hold, modern tourists will look for emotional connections and deeper contacts with the local culture and value the interaction with the residents in the tourism process [4]. The outcome of this interaction plays an important role not only in terms of tourists’ consumption and satisfaction with the destination but also in terms of residents’ attitudes [5]. In fact, the resident–tourist interaction also plays an important role in the perception of the behavior of the residents of the destination. In conjunction with higher tourist satisfaction, residents become more supportive of local tourism development [6]. Residents’ support or lack of support for tourism development as a unique value presence of the destination will also play an important role for the brand ambassador. Therefore, in the context of the continued development of the COVID-19 pandemic, how does the ability of residents of tourism destinations to be psychologically resilient affect their attitudes and behavioral intentions toward tourism development? What are the factors that promote psychological resilience and resident–tourist interaction? To better answer these questions, this study constructs a moderated mediation model that includes psychological resilience, resident–tourist interaction, cognitive reappraisal, and brand ambassador behavioral intention, and examines not only the relationship between the recovery of residents’ psychological resilience under the influence of the COVID-19 pandemic and destination brand ambassador behavioral intention, but also the factors that moderate residents’ psychological resilience. In the context of tourism destination management practice, the results of this study can provide a theoretical basis and a practical reference for cultivating residents to become destination brand ambassadors in the context of the COVID-19 pandemic.

## 2. Literature Review and Hypothesis Development

### 2.1. Psychological Resilience

Carpenter et al. defined resilience as a multifaceted and multidisciplinary field of study that seeks to explore how research subjects respond to change [7]. In the thematic research on the construction and development of tourism destinations in the context of the COVID-19 pandemic, resilience theory can provide solution strategies. For example, Traskevich and Fontanari integrated a conceptual model of destination resilience based on a literature review in the context of the COVID-19 pandemic, conducted research on more than 1000 German tourism companies to confirm the validity of a resilience-oriented market framework [8]. Bertella also used the COVID-19 pandemic as a context to analyze the correlation between resilience and multidimensional sustainability using a qualitative research approach with tourism community members [9]. Ngoc et al. used the COVID-19 pandemic as a context to present an empirical study in Vietnamese tourism enterprises, it was proposed that human resource construction and practices can improve the resilience of employees in response to the epidemic in order to improve organizational resilience [10]. Prayag studied the factors influencing tourism resilience at three levels and pointed out that the resilience of four groups, namely, residents of tourism destinations, employees of tourism enterprises, tourists, and other tourism-related temporary populations, constitutes the micro-level of tourism system resilience, and emphasized that the quality of resident–tourist interaction in a community depends to a large extent on the attitudes and behaviors of residents [11]. In general, the destination, community, and organizational levels of resilience management cover a more comprehensive range, but none of them can be separated from the foundation of the psychological resilience of destination residents.

Psychological resilience belongs to the study of positive psychology, which emphasizes humanistic thinking in the field of tourism, coinciding with the tourism industry’s pursuit of increasing people’s well-being, and its application in the field of tourism is mostly focused on the tourist experience and less on the residents and other stakeholders of the destination. Regarding the conceptual definition of psychological resilience, the American Psychological Association defined psychological resilience referring to the thoughts and behaviors that individuals learn and develop as they recover from and adapt to adversity, threats, or stress [12]; Connor and Davidson considered psychological resilience as the ability to help individuals successfully recover from adversity or stress [13]; Ungar proposed from a process theory perspective that psychological resilience is a process in which individuals interact with their environment and that this process requires the role of factors such as individual traits (i.e., temperament, personality) and social factors [14]. Regarding theoretical models of psychological resilience mechanisms, Kumpfer argued that the simple “factor labels” of “resilience factors” or individual protective traits mask the complex interactions between psychologically resilient individuals and their environment, and that psychologically resilient individuals actively create their own environment [15]; the psychological resilience model constructed by Richardson describes the temporary equilibrium of a person’s physical, mental, and spiritual adaptation to the external environment at a certain point in time, which is influenced by the interaction of various protective and risk factors from inside and outside the individual [16].

Positive organizational behavior builds on the findings of positive psychology, but differs from it in that it focuses on mental states that can be changed. Luthans defined it as an applied discipline that measures, develops, and effectively manages mental abilities for the purpose of improving work performance, and is oriented toward positive employee dynamics [17]. Psychological capital is defined as the state of positive psychological development of a person and when applied to the workplace, is known as positive organizational behavior. According to positive organizational behavior, resilience is an important feature of psychological capital, and is also considered to be of great value in corporate development, employee resilience has significant value in business development, that the strength of an employee’s resilience determines his success or failure to a greater degree than education, experience, or training, and that employee resilience is positively related to job-related outcomes [18]. Although there is no clear boundary between employees’ psychological resilience in particular and employee resilience in general, it has been verified that individuals with high psychological resilience have higher employee resilience in tourism organizations [19].

### 2.2. Brand Ambassador Behavioral Intentions

According to Aaker’s customer-based brand equity (CBBE) model, brand equity measures are divided into five dimensions: awareness, associations/image, perceived quality, loyalty, and brand assets [20]. Considering the difference between service brand and product brand, Boo et al. proposed to measure destination brand equity in terms of destination brand awareness (DBA), destination brand image (DBI), destination brand quality (DBQ), destination brand loyalty (DBL), and destination brand value (DBV) [21]. Govers argued that branding by establishing a brand that reflects the local cultural identity and creating conditions for residents to become local brand ambassadors is an important branding tool in order to achieve a non-marketing approach (i.e., internal branding) to positioning the brand [22]; as Braun et al. argued, residents have an important position as internal stakeholders, playing four distinct but concurrent roles as marketing campaign audiences, interactors with tourists, brand ambassadors, and legal citizens [23]. The concept of internal marketing was first introduced by Grönroos, whose core idea is that companies should focus on internal customers and internal service offerings and use sound internal marketing strategies to improve efficiency and thereby build overall competitive advantage for the company [24]. Sheehan and Ritchie combine internal marketing with destination branding and emphasize that internal stakeholder commitment to a destination brand can often be facilitated more through communication links than contractual links [25]; Sartori et al. suggested that the success of a brand strategy depends on the willingness of its internal stakeholders to support marketing efforts, and that the assessment of overall brand equity from the internal side is equally important, and propose an internal stakeholder-based model of brand equity for tourism destinations [26].

On the one hand, residents’ participation in destination brand development is considered as a brand-building behavior. Chen et al. argued that residents’ participation in tourism development is crucial because of the increasing number of interactive behaviors in tourism activities, making the brand-building behavior consist of four dimensions: participation, word-of-mouth promotion, ambassador behavior, and retention through literature review [27]. On the other hand, companies see their employees as brand ambassadors who translate the brand vision into brand reality [28]. Andersson and Ekman proposed the concept of ambassador networks that include residents as an effective communication tool to enhance the competitiveness of tourism destinations [29]; Cai suggested that destination brand ambassador behavior can be summarized as an effective tool for building stronger destination brand identity [30]; Wassler defined destination brand ambassador behavior as planned or spontaneous destination brand promotion-related or development-related behavior designed to increase equity in destination branding and classifies it into five dimensions: word-of-mouth promotion, online word-of-mouth promotion, participation in future brand-related promotion activities, participation in future brand development, and personal use of brand promotional materials [31].

### 2.3. Resident–Tourist Interaction

Homans first proposed social exchange theory, emphasizing that it is a basic theory used to study human behavior based on sociology, psychology, and economics [32]. Perdue et al. introduced social exchange theory into tourism research, arguing that destination residents determine their attitudes and behaviors toward tourism development by estimating the difference between the costs they pay and the economic benefits they receive from the process [33]. Nunkoo used social exchange theory to study the attitudes of destination residents towards tourism, showing that the value of the economic, social, and cultural elements of the resident–tourist exchange process influence the way residents perceive tourism development and determine the degree of acceptance of it [34]. In addition, Sutton made an early reference to the concept of resident–tourist interaction and argued that interaction between destination residents and tourists may provide opportunities for communication and may also reinforce residents’ urge to exploit [35]. Some scholars also referred to the relationship between destination residents and tourists in tourism and explored the impact of tourism development on destination communities in 1977 [36].

Tourism activities cannot be separated from social interaction, residents and tourists as important subjects of tourism activities generated by different degrees of interactive behavior is called the resident–tourist interaction [37]. From the perspective of the residents, the resident–tourist interaction has an impact on the economic, sociocultural, and environmental changes in the destination residents’ perception of tourism development [38], on the attitude of destination residents towards tourism development [39], and on the quality of life of the destination residents [40]. From the perspective of the tourists, the resident–tourist interaction affects the tourist’s perception of the image of the destination [41], the tourist’s experience of tourism [42].

### 2.4. Cognitive Reappraisal

Research on emotion regulation originated in the field of psychology, where Thompson argued that rather than limiting the intake of emotionally evocative information, individuals self-regulate their emotions by changing their interpretation of this information, and that emotion regulation strategies can significantly influence the course of social interactions and the development of social relationships [43]. Gross defined emotion regulation as the process by which individuals exert influence over what emotions they have, when they have them, and how they experience and express them [44]. According to the definition of emotion regulation, two types of emotion regulation can be distinguished, the prior focus strategy and the response focus strategy, cognitive reappraisal is a focus strategy in emotion regulation, which is a form of cognitive adjustment that alters emotions by changing the individual’s interpretation of potential emotion-triggering situations [45]. Cognitive reappraisal facilitates the differentiation of positive affective states that do not require change from negative states that require regulation, thereby increasing positive emotions and decreasing negative emotions [46]. Moreover, individuals who aim to increase positive emotions have a greater effect on cognitive reappraisal success compared to individuals who aim to decrease negative emotions [47]. Individuals who use cognitive reappraisal strategies to regulate emotions tend to have better affective patterns, social relationships, mental health, and well-being [48].

Paying attention to the way employees in the organization regulate their emotions is necessary because emotional dysregulation increases turnover intentions and may manifest itself only in job dissatisfaction, which may manifest itself in dissatisfaction with the job and reduced organizational commitment [49]. Schraub et al. suggested that emotion regulation is a process of recovery from work stress and that individual differences in emotion regulation influence work-related emotional events and performance styles [50]. Semmer et al. found that in the workplace, employees’ adoption of regulation that inhibits negative emotions has a positive effect on the quality of their perceived interactions [51]. Vîrgă demonstrated that cognitive reappraisal acts as a buffer between job insecurity and work engagement, and that training aimed at improving emotion regulation skills can help increase employee engagement at work [52].

### 2.5. Hypothesis Development

Firstly, resilient individuals tend to actively prepare for difficulties and minimize the impact of stressful events on themselves by using their psychological resources effectively [53]. Particularly during the COVID-19 pandemic, Killgore et al. found that the level of self-perceived psychological resilience of the U.S. population was generally adversely affected by the ongoing crisis, with those with lower psychological resilience reporting difficulty coping with the emotional challenges of a pandemic crisis [54]; Song et al. showed through their study that individuals with higher psychological resilience and positive coping styles had lower levels of anxiety and depression [55]. Residents, as tourism practitioners, have not received more attention to their psychological resilience under the stresses brought about by the COVID-19 pandemic. This study considers psychological resilience as the ability to self-perceive traits in the face of adversity and concludes that destination residents with high psychological resilience are better able to adapt to the changes brought about by the COVID-19 pandemic.

Secondly, the organization-centered perspective proposed by Luthans and the employee-centered perspective proposed by Wright constitute a win-win situation for both organizations and employees [56]. Bakker and Schaufeli emphasized that employee involvement increases our understanding of positive organizational processes in organizations, which would be a focus of research [57]. Residents play an important role in participating in destination branding, guided by positive organizational behavior, and individual differences in psychological resilience influence the process by which individuals strive to confront and transcend crises and actively pursue new experiences. Specifically, psychological resilience plays a buffering role in stress and participation in brand-building behavior. Individual residents of destinations with higher psychological resilience are better able to face the negative effects of the COVID-19 pandemic and are more willing to contribute their efforts to play a facilitating role in the process of destination brand building and have a higher intention to act as brand ambassadors. In addition, Prayag emphasized the need for research on destination resilience, psychological resilience, employee resilience, and organizational resilience, and argued that the dynamic relationship between residents’ high psychological resilience characteristics and interactions with tourists has yet to be verified [58]. Based on the arguments above, Hypotheses 1 and 2 for this study are presented:

**Hypothesis** **1.**
*The psychological resilience of residents of tourism destinations positively influences their brand ambassador behavioral intentions.*


**Hypothesis** **2.**
*The psychological resilience of residents of tourism destinations positively influences resident–tourist interaction.*


As internal stakeholders of the destination, residents participate in its branding as a reflection of their internalization of that brand. Residents are not only important marketers of destination brands, but also service providers who have direct contact with tourists and transform marketing communication brands into actual tourist experience brands through their own tourism-related services as well as their contact with tourists. Gowreesunkar analyzed how residents fulfill the brand promise of the destination marketing organization (DMO) through their interactive behavior with tourists [59]. Weaver et al. emphasized the citizenship of destination residents, through their hospitality and collaborative engagement in promoting the sustainable development of the destination [60]. San et al. suggested that destination residents play such an important role in branding probably because of their positive interactive behaviors with tourists, presenting a positive destination brand image to existing or potential tourists [61]. In general, the willingness of residents to interact with and provide good services to tourists is key and a prerequisite to the communication of destination branding. In addition, Palmer et al. suggested that residents with positive attitudes towards tourism play a crucial role in tourism participation as community or destination ambassadors [62]. Residents are considered to have the ability to develop and promote the destination brand and eventually become destination brand ambassadors [63]. Based on the arguments above, Hypotheses 3 and 4 for this study are presented:

**Hypothesis** **3.**
*Resident–tourist interaction positively influences the brand ambassador behavioral intentions of residents of tourism destinations.*


**Hypothesis** **4.**
*Resident–tourist interaction plays a mediating role in the relationship between psychological resilience and brand ambassador behavioral intentions.*


Tugade et al., analyzed the mechanism of action of psychological resilience through the use of positive emotions to cope with rebound from negative events [64]. Kumpfer confirmed that individuals with high psychological resilience have positive emotional traits [15]. In organizational settings, employees with high psychological resilience experience more positive emotions than those with low psychological resilience and view the process and outcomes of change more optimistically, and their positive emotions moderate the relationship between their psychological resilience and their attitudes [65]. Both psychological resilience and cognitive reappraisal are important variables that are evaluated in psychological terms, and both are related to positive emotions in organizational behavior such that positive emotional responses further influence employees’ attitudes and behaviors. It is reasonable to positively influence the quality of interactions by showing positive and suppressing negative emotions because emotions are contagious [66]. To gain insight into whether cognitive reappraisal has a facilitative effect in the degree to which psychological resilience influences brand ambassadors’ behavioral intentions, this study sets cognitive reappraisal as an intervention variable to investigate whether the relationship between psychological resilience and resident–tourist interaction changes as a result of cognitive reappraisal. We also intend to determine whether the mediating effect of resident–tourist interaction holds after cognitive reappraisal as well as whether its mediating role changes. Based on the arguments above, Hypotheses 5 and 6 for this study are presented:

**Hypothesis** **5.**
*Cognitive reappraisal plays a positive moderating role in the relationship between psychological resilience and resident–tourist interaction.*


**Hypothesis** **6.**
*Cognitive reappraisal positively moderates the mediating role of resident–tourist interaction in the relationship between psychological resilience and brand ambassador behavioral intentions.*


Based on the above analysis, we construct a moderated mediation model to explore the effects of residents of tourism destinations’ psychological resilience on their brand ambassador behavior. First, residents of tourist destinations with high psychological resilience have a higher degree of adaptation to pandemic-induced changes and can directly control their attitudes and behavioral intentions to participate in destination brand development (i.e., their attitude toward resident–tourist interaction and behavioral intentions toward brand ambassadorship). Second, residents’ participation in destination brand development cannot be separated from resident–tourist interaction, which has a direct impact on destination brand ambassador behavior, and residents’ psychological resilience has an indirect impact on their brand ambassador behavioral intentions through their attitude toward resident–tourist interaction. Finally, the emotion regulation strategy of cognitive reappraisal, as a moderating variable, has a facilitating effect on the relationship between residents’ psychological resilience and their attitudes toward resident–tourist interaction. The mediating effect of resident–tourist interaction still holds under the moderating effect of cognitive reappraisal, and the mediating effect of resident–tourist interaction is enhanced under high levels of cognitive reappraisal. The hypothesis and conceptual framework of this study (Figure 1) are as follows.

## 3. Materials and Methods

### 3.1. Sample

The research site of this study was the Guyan Picture Town Scenic Area (thereafter referred as Guyan) in Lishui City, Zhejiang Province, China. It is about 94 square kilometers with about 12,000 residents in 2018. There are two unique local attractions that support Guyan’s destination brand: the Tongji Qu (an ancient irrigation channel built in 505 AD) and the Babisong Painting style that originated in the region. In 2014, Guyan was designated as a 4A National Tourist Attraction (the highest grade in China is 5A) [67,68]. In 2018, the number of visits increased to about 1.77 million, demonstrating significant tourism growth [69]. Tourism income in terms of admissions grew from 1.45 million RMB (about 217 thousand USD) in 2012 to about 22 million RMB (about 3.3 million USD) in 2018, further reflecting this significant growth. Therefore, it is representative to choose Guyan Picture Town as the case study place, and the main object of the study is the residents who are engaged in tourism business activities there, mainly involving souvenir stores, scenic spots, farmhouses, and so on. Questionnaires were distributed offline. In the pre-study stage, 20 initial questionnaires were distributed to collect residents’ opinions on the design of the questions, and 20 valid questionnaires were collected, with a valid recovery rate of 100%. In the formal research stage, 200 questionnaires were distributed and 174 valid questionnaires were collected after excluding incomplete and invalid questionnaires, for a valid recovery rate of 87%. A total of 220 questionnaires were distributed in the pre-research and formal research stages and 194 valid questionnaires were collected, for a valid recovery rate of 88%. The whole process is divided into two stages and lasted for 14 days. The interviewing time was concentrated on lunch breaks and evening closing times so as not to disturb normal business.

Among the 194 valid samples, 115 (59.3%) were filled out by women and 79 (40.7%) by men. The average age of the respondents was 44 years, and in terms of age distribution, 11 (5.7%) respondents were between 20 and 30 years old, 69 (35.6%) were between 30 and 40 years old, 59 (30.4%) were between 40 and 50 years old, and 55 (28.4%) were between 50 and 60 years old. In terms of education level, no respondents had an elementary school education or below, 36.6% had a junior high school education, 22.7% had high school or junior college, 28.4% had college or high school, 11.9% had a bachelor’s degree, and 0.5% had a graduate degree or higher. In addition, 188 (96.9%) respondents were married and only 6 (3.1%) were unmarried. The monthly personal income of the respondents was concentrated in the range of RMB 2000 to RMB 8000 (97.9%), with only 2 respondents having a monthly personal income of less than RMB 2000 and 2 respondents having a monthly personal income of more than RMB 8000.

### 3.2. Variables and Instruments

The four variables in the conceptual model (psychological resilience, resident–tourist interaction, brand ambassador behavioral intentions, and cognitive reappraisal) were measured using validated measurement scales, and the initial questionnaire was revised in the pre-research phase in conjunction with the research needs to form the survey questionnaire used in this study.

Residents’ psychological resilience was measured using the Connor–Davidson Resilience Scale (CD-RISC), which contains 25 items [13]. Campbell-Sills and Stein revised it to form a 10-item unidimensional measurement scale for psychological resilience [70]. In this study, the 10-item items were (1) I am adaptable to change; (2) I can handle anything; (3) I can see the positive side of things; (4) Coping with stress makes me stronger; (5) I tend to recover after illness or difficulty; (6) I can face difficulties and work hard to achieve goals; (7) I can stay focused under stress; (8) I am not easily discouraged by failure; (9) I consider myself a strong person; and (10) I can handle unpleasant feelings. Question item ratings were assessed using a seven-point Likert scale (0 = strongly disagree, 7 = strongly agree), which had a Cronbach’s alpha coefficient of 0.89 and high internal consistency.

Residents’ attitudes toward resident–tourist interaction were applied to the question items developed by Teye et al. in their study of residents’ attitudes toward tourism development [71]. Three of these question items were selected for this study, which were (1) I have developed friendships with tourists, (2) My interactions with tourists are positive and helpful, and (3) I enjoy interacting with tourists. The question item ratings were assessed using a seven-point Likert scale (0 = strongly disagree, 7 = strongly agree), which had a Cronbach’s alpha coefficient of 0.86 and high internal consistency.

The residents’ behavioral intentions to be a brand ambassador for their destination were measured using the brand ambassador behavioral intentions (BABI) scale developed by Wassler et al. [72]. In this study, seven items were selected, which were (1) If I had the opportunity, I would write articles about Guyan Picture Town online so that more people would know about it; (2) If I had the opportunity, I would send Guyan Picture Town to my friends online; (3) If I have the opportunity, I will promote Guyan Picture Town as much as possible on the Internet to ensure that more people know about it; (4) I plan to participate in future promotional activities (such as festivals and exhibitions); (5) If I have the opportunity, I will make my own contribution to the development of Guyan Picture Town (such as following and participating in events); (6) I plan to participate in the future development of Guyan Picture Town (e.g., pay attention to and participate in activities); (7) If I have the opportunity, I will use the promotional materials for Guyan Picture Town. Question item ratings were assessed using a seven-point Likert scale (0 = strongly disagree, 7 = strongly agree), which had a Cronbach’s alpha coefficient of 0.90 and a high degree of internal consistency.

The residents’ cognitive reappraisal strategy was applied using the Emotion Regulation Questionnaire (ERQ) developed by Gross and John [44]. Three of these items were selected for this study, namely, (1) I control my emotions by changing my perceptions of my current situation, (2) I change the way I think about my current situation when I want to feel fewer negative emotions (e.g., sadness or anger), and (3) I change the way I think about my current situation when I want to feel more positive emotions (e.g., happiness or pleasure). Question item ratings were assessed using a seven-point Likert scale (0 = strongly disagree, 7 = strongly agree), which had a Cronbach’s alpha coefficient of 0.85 and high internal consistency.

### 3.3. Statistical Analysis

The data were analyzed mainly using SPSS (version 22.0, IBM Corp., Armonk, NY, USA) and Mplus (version 7.4, Muthén & Muthén, Los Angeles, CA, USA). Descriptive statistical and correlation analyses were performed using SPSS software, with Cronbach’s alpha coefficient as the criterion for reliability testing; validation factor analysis was performed using Mplus software to test the differential validity of each variable, latent variable model estimation was performed using the great likelihood method, and the model paths were subjected to a bootstrap test for bias correction, set with 1000 repetitions of sampling times and 95% confidence intervals.

## 4. Results

### 4.1. Reliability and Validity

The results of the reliability test and validation factor analysis are shown in Table 1, where the composite reliability (CR) of the four variables ranges from 0.87–0.92, the Cronbach’s alpha coefficient ranges from 0.86–0.90, and the average variance extracted (AVE) is greater than 0.5 for all four variables, and thus the reliability and convergent validity of the four variables are supported [73,74,75]. In addition, the AVE values of each variable are higher than the respective inter-variate correlations, thus showing differential validity [74,75].

### 4.2. Structural Equation Modeling (SEM) Analysis

#### 4.2.1. Testing the Direct and Mediating Effects

First, in the bivariate model of psychological resilience and brand ambassador behavioral intentions, there is a significant positive effect of psychological resilience on brand ambassador behavioral intentions (β = 0.650, *p* < 0.001), thus Hypothesis 1 is supported, indicating that residents with higher psychological resilience scores are more willing to become destination brand ambassadors and participate in destination branding. Second, hypothesis testing of the three variables model of psychological resilience, resident–tourist interaction, and brand ambassador behavioral intentions have better model fit data χ^2^ = 286.925 (df = 167, *p* < 0.001), χ^2^/df = 1.718, comparative fit index (CFI) = 0.907, Tucker–Lewis index (TLI) = 0.894, root mean square error of approximation (RMSEA) = 0.086, and standardized root mean square residual (SRMR) = 0.065 [76]. The standardized path coefficients and hypothesis testing results are shown in Table 2, where psychological resilience has a significant positive effect on resident–tourist interaction (β = 0.700, *p* < 0.001), and Hypothesis 2 is verified, indicating that residents with higher psychological resilience scores are more active in interacting with tourists and enjoy the fun and help that comes from interaction with tourists more. It can therefore be seen that there is a significant positive effect of resident–tourist interaction on brand ambassador behavioral intentions (β = 0.613, *p* < 0.001), Hypothesis 3 is supported, indicating that residents’ attitudes and perceptions of interactions with tourists influence their participation in destination branding. The effect of psychological resilience on brand ambassador behavioral intentions is not significant (*p* > 0.05). The above results indicate that resident–tourist interaction plays a fully mediating role in the relationship between psychological resilience and brand ambassador behavioral intentions, specifically, residents with high psychological resilience support destination brand development, and this process is achieved through interaction with tourists. Further, the mediating effect of resident–tourist interaction was tested with reference to MacKinnon et al.’s study [77]. The results are shown in Table 3, where the mediation effect is 0.429 (*p* < 0.001) with a 95% bootstrap confidence interval of [0.279, 0.881], which does not contain 0. Hypothesis 4 is thus supported.

#### 4.2.2. Testing the Moderating and Moderated Mediation Effects

Referring to Hayes’ study, the mediating and moderating effects were included in the same model for integrated analysis with adjustment [78]. To avoid the multicollinearity problem, the variables were first centralized when constructing the interaction variable. The variables of cognitive reappraisal and (psychological resilience * cognitive reappraisal) were added for hypothesis testing, and according to the latent moderated structural equations (LMS) method, the akaike information criterion (AIC) and H0 indicators of Model 1 after adding the interaction were smaller than those of Model 0 (i.e., without the interaction), the results are shown in Table 4 and indicate a better model fit [79]. The standardized path coefficients and hypothesis testing results are shown in Table 5, and the interaction variables of psychological resilience and cognitive reappraisal have a significant positive effect on resident–tourist interaction (β = 0.385, *p* < 0.05), thus indicating that cognitive reappraisal moderates the effect of psychological resilience on resident–tourist interaction. Hypothesis 5 is therefore verified, indicating that the degree of influence of psychological resilience on the resident–tourist interaction changes under the intervention of cognitive reappraisal, residents with high psychological resilience produce higher positive emotions and have more positive attitudes towards the resident–tourist interaction when using the emotion regulation approach of cognitive reappraisal.

To test the moderated mediating effect, the effect of residents’ psychological resilience on brand ambassador behavioral intentions through resident–tourist interaction was calculated at different levels of cognitive reappraisal (e.g., mean plus or minus one standard deviation), and the results are shown in Table 6, with a simple effect test and the interaction effect plotted in Figure 2, showing that the emotion regulation strategy of cognitive reappraisal can enhance the facilitative effect of psychological resilience on the resident–tourist interaction. The moderating effect of cognitive reappraisal on the psychological resilience—resident—tourist interaction—brand ambassador behavioral intentions mediating mechanism was significant (moderating mediating effect index = 0.211, *p* < 0.05). The 95% bootstrap confidence interval was [−0.142, 0.569] at the cognitive reappraisal level of the low group, which included 0, and the mediating effect of resident–tourist interaction was not significant; the 95% bootstrap confidence interval was [0.196, 0.739] and [0.279, 1.103] at the cognitive reappraisal level of the medium and high group, respectively, which did not contain 0, at which time the mediating effect of resident–tourist interaction is significant; in addition, the indirect effect index under the cognitive reappraisal level of the high grouping is greater than that under the cognitive reappraisal level of the medium grouping, which indicates that the mediating effect under the cognitive reappraisal level of the high grouping is stronger. The above analysis shows that cognitive reappraisal positively moderates the mediating effect of resident–tourist interaction between psychological resilience and brand ambassador behavioral intentions, and the higher the level of cognitive reappraisal, the stronger the mediating effect of resident–tourist interaction is. Hypothesis 6 is therefore verified, indicating that residents with high psychological resilience are more willing to present a brand image by interacting with tourists and acting as brand ambassadors when a higher level of emotional regulation strategy of cognitive reappraisal is used.

## 5. Discussion

This study investigates the mechanism of residents’ psychological resilience on their behavioral intentions to be brand ambassadors in the context of the COVID-19 pandemic, using residents of tourist destinations as the research target. Focusing on the mediating and moderating effect mechanisms of two variables, resident–tourist interaction and cognitive reappraisal, the results of the empirical tests show that: (1) residents’ psychological resilience has a positive effect on brand ambassador behavioral intentions; (2) the direct effect of psychological resilience on them is not significant after adding the mediating variable of resident–tourist interaction, which indirectly affects brand ambassador behavioral intentions through resident–tourist interaction; (3) cognitive reappraisal has a facilitative effect on the relationship between psychological resilience and resident–tourist interaction, and the mediating effect is significant at high and medium levels of cognitive reappraisal and not significant at low levels; the higher the level of cognitive reappraisal, the stronger the mediating effect of resident–tourist interaction is.

The theoretical model and findings constructed in this study have theoretical implications for research on organizational behavior and branding aspects of destinations.

First, this study focuses on the psychological resilience of a population of residents in tourism destinations, expanding the research on psychological resilience in applied populations. Previous studies have focused on adolescents, physicians, the sick, and the elderly facing abuse, major trauma, and stress [80,81,82], exploring the impact of psychological resilience on health, and less on tourism, failing to reveal the mechanisms influencing the psychological resilience of residents in tourist destinations. Based on previous studies in the literature, this study argues that psychological resilience reflects the ability of residents to recover after an epidemic shock, and the results of the empirical analysis confirm that residents with high psychological resilience are able to actively cope with restricted business activities and other changes caused by the epidemic [65]. Similarly, residents with high psychological resilience are more likely to engage in good interactions with visitors and participate in word-of-mouth marketing to demonstrate their image as hosts in business activities.

Second, there is a theoretical and practical basis for research on destination brand ambassador behavior, which has enriched the scope of research on the antecedent influences on brand ambassador behavior. Currently, in the field of internal brand management research, the issue of brand ambassadorship is mostly studied at the level of corporate members and consumers, who are believed to be motivated to exhibit brand ambassadorial behavior only after they truly understand the brand connotation and establish some relevance to the brand [83]. Destination brand ambassador behavior, on the other hand, is targeted at community residents as an important means of participation in destination branding, where residents’ participation in brand promotion efforts will lead to increased ownership of the brand and thus a greater sense of responsibility for its development and management of brand image [23]. Because of this, this study discusses the factors that influence brand ambassadorial behavior, effectively complementing research in the area of brand ambassadorial behavior in tourism destinations.

Third, this study effectively extends the findings of resident–tourist interaction by introducing it into the relationship between psychological resilience and destination brand ambassadorship. Based on social exchange theory, previous studies have shown that “resident–tourist interaction” is not only the most important factor affecting tourist satisfaction, but also an important factor affecting residents’ support for tourism [6,84]. However, this study goes deeper and suggests that psychological resilience will have a positive impact on resident–tourist interactions, while the degree and quality of resident–tourist interactions will further promote the performance of residents as brand ambassadors, enriching the existing research on resident–tourist interactions.

Fourth, this study explored the role of cognitive reappraisal, which provides an example of the restoration of psychological resilience and the quality of resident–tourist interaction in destination residents. Previous studies have dealt more with psychological aspects, exploring the effects of cognitive reappraisal on depression and anxiety disorders [85]. Similar to psychology, which explores their dynamic effects on treatment outcomes as mediating variables, psychological resilience and cognitive reappraisal are directly related to residents’ positive emotions, which in turn have an impact in terms of their attitudes and behavioral intentions towards tourism development [48,64,86]. In this study, cognitive reappraisal was set as a moderator and confirmed the influence of cognitive reappraisal on the “psychological resilience-resident–tourist interaction-brand ambassador behavioral intention” path. When residents with high psychological resilience used the emotion regulation strategy of cognitive reappraisal, they had more supportive attitudes toward the resident–tourist interaction. At the same time, the positive effect of psychological resilience and brand ambassador behavioral intentions, which were mediated through the resident–tourist interaction, was enhanced with the support of high levels of cognitive reappraisal. The results also suggest that the residents’ reappraisal and regulation of emotions help them to cope and achieve tourism-related tasks. This, to some extent, also bridges the gap of cognitive reappraisal in tourism destination field research.

## 6. Conclusions

Focusing on the group of residents in tourist destinations affected by the COVID-19 pandemic, this study explores the psychological resilience traits of residents and their attitudes and behavioral intentions towards participation in destination branding. Firstly, this study uses positive organizational behavior as a guiding principle, combines with destination branding, and confirms that residents with high psychological resilience have more enthusiastic resident–tourist interaction and more positive brand ambassador intentions, and that residents tend to be more willing to show and promote the destination brand by generating interactions with tourists. Secondly, this study introduces the emotion regulation strategy of cognitive reappraisal and argues that residents who adopt this strategy are able to generate higher positive emotions, synergizing with positive emotions generated by psychological resilience and promoting work engagement and supportive attitudes towards tourism development among residents with high psychological resilience in destination brand development. Finally, the emotion regulation strategy of cognitive reappraisal stimulates resident–tourist interaction and emphasizes the importance of resident–tourist interaction as a prerequisite for being destination brand ambassadors.

Based on the above findings, this study proposes the following management implications. First, destination managers should give more care and support to residents facing stress, not just stress due to the COVID-19 pandemic limiting tourism development, and helping residents recover from adversity can increase their supportive attitudes towards tourism development. Second, regarding residents as the role of the tourism process to provide services to tourists, the emotion problems in the work process are worth drawing attention to, especially in the context of the COVID-19 pandemic and fear of the spread of disease, timely psychological guidance and emotion regulation can ease the resistance of residents to tourism development. Third, destination managers should pay attention to the brand ambassador image of residents, which can help a lot in shaping the destination brand image, and residents can improve tourist satisfaction and loyalty through good resident–tourist interaction, while there is a higher demand for encouraging residents to participate in destination branding activities.

This study has certain limitations. First, the research site was chosen in the Guyan Picture Town scenic area in Lishui City, Zhejiang Province, China, which is symbolic but should be further studied in the future to consider the characteristics of various types of tourist destinations. Second, the research design used in this application is cross-sectional, and it is therefore difficult to draw more accurate causal conclusions, which can be explored more deeply in future research through the use of tracking studies. Finally, this study only considered the moderating effect of cognitive reappraisal and did not discuss the intervention mechanisms of other emotion regulation strategies, thus differences in the moderating effects of various emotion regulation strategies can be explored in the future.

## Figures and Tables

**Figure 1 behavsci-12-00337-f001:**
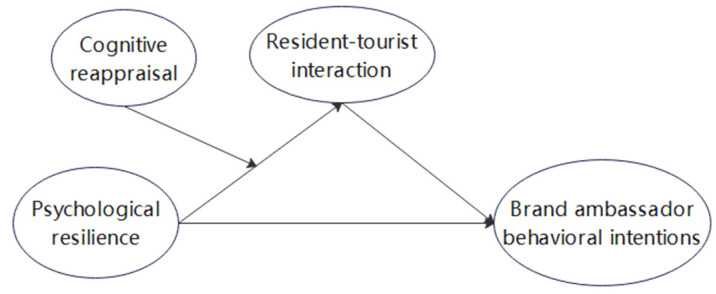
Conceptual framework.

**Figure 2 behavsci-12-00337-f002:**
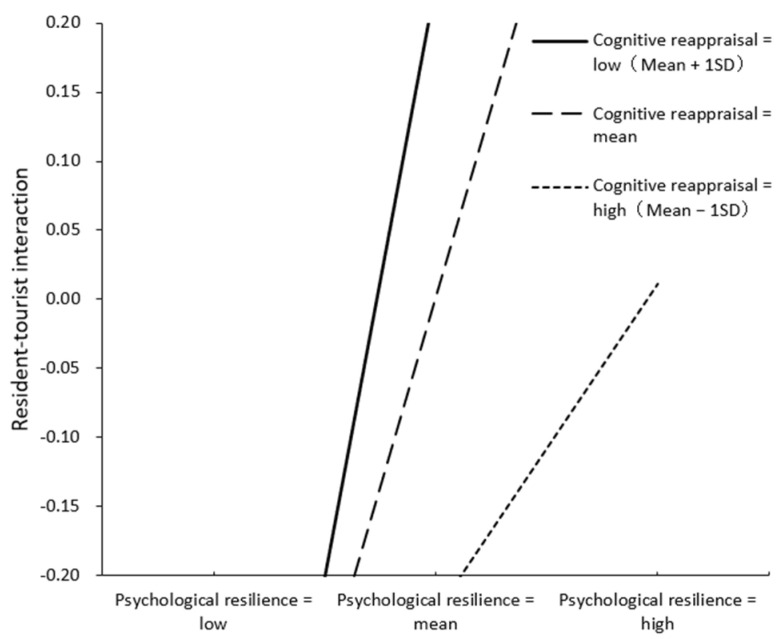
The moderating effect of cognitive reappraisal on psychological resilience and resident–tourist interaction.

**Table 1 behavsci-12-00337-t001:** CR, AVE, and correlation matrix.

	Cronbach’s α	CR	AVE	1	2	3	4
1. Psychological resilience	0.891	0.914	0.516	**0.718**			
2. Resident–tourist interaction	0.865	0.872	0.694	0.698	**0.833**		
3. Cognitive reappraisal	0.855	0.918	0.790	0.379	0.422	**0.889**	
4. Brand ambassador behavioral intentions	0.902	0.920	0.623	0.654	0.770	0.510	**0.789**

Note. CR refers to composite reliability. AVE refers to average variance extracted. The square root of each corresponding AVE score is shown in bold.

**Table 2 behavsci-12-00337-t002:** Direct effects.

Hypothesized Paths	Factor Loading	SD	t	*p*	Results
Psychological resilience → resident–tourist interaction	0.700	0.066	10.542	0.000	Supported
Psychological resilience → brand ambassador behavioral intentions	0.225	0.118	1.900	0.057	Not supported
Resident–tourist interaction → brand ambassador behavioral intentions	0.613	0.111	5.548	0.000	Supported

**Table 3 behavsci-12-00337-t003:** Mediating effect.

Hypothesized Paths	95% CI		*p*	Results
	LLCI	ULCI		
Psychological resilience → resident–tourist interaction → brand ambassador behavioral intentions	0.279	0.881	0.000	Supported

Note. CI refers to confidence interval. LLCI refers to the lower level of the 95% confidence interval. ULCI refers to the upper level of the 95% confidence interval.

**Table 4 behavsci-12-00337-t004:** Moderated mediation model.

Indicator	Model 0	Model 1	Analysis (Model 0—Model 1)	Results
H0	−2114.654	−2111.586	0.013 > 0	Supported
AIC	4377.309	4373.172	4.137 > 0	Supported
Df	74	75		
X²/Df	1.650			
CFI	0.907			
TLI	0.896			
RMSEA	0.081			
SRMR	0.078			

Note. Model 0 refers to the model without the interaction variable of (psychological resilience * cognitive reappraisal). Model 1 refers to the model that includes the interaction variable of (psychological resilience * cognitive reappraisal).

**Table 5 behavsci-12-00337-t005:** Direct and indirect effects (model 1).

Hypothesized Paths	Factor Loading	SD	t	*p*	Results
Psychological resilience → resident–tourist interaction	0.683	0.188	3.631	0.000	Supported
Cognitive reappraisal → resident–tourist interaction	0.228	0.142	1.609	0.108	Not supported
(Psychological resilience * cognitive reappraisal) → resident–tourist interaction	0.385	0.195	1.972	0.049	Supported
Psychological resilience → brand ambassador behavioral intentions	0.272	0.178	1.529	0.126	Not supported
Resident–tourist interaction → brand ambassador behavioral intentions	0.548	0.133	4.133	0.000	Supported

Note. Model 1 refers to the model that includes the interaction variable of (psychological resilience * cognitive reappraisal).

**Table 6 behavsci-12-00337-t006:** Moderated mediation effect.

	Indirect Effect/Index	SD	t	95% CI		*p*	Results
				LLCI	ULCI		
Mean − 1SD	0.147	0.176	0.835	−0.142	0.569	0.404	Not supported
Mean	0.375	0.133	2.824	0.196	0.739	0.005	Supported
Mean + 1SD	0.603	0.174	3.455	0.279	1.103	0.001	Supported
Moderated mediation effect	0.211	0.106	1.997	0.071	0.559	0.046	Supported

Note. SD refers to standard deviation. LLCI refers to lower level of the 95% confidence interval. ULCI refers to upper level of the 95% confidence interval.

## Data Availability

The data analyzed in this paper are proprietary, and therefore cannot be posted online.

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
