# Peer review of "Exploring How the Psychological Resilience of Residents of Tourism Destinations Affected Brand Ambassador Behavior during the COVID-19 Pandemic"

_behavsci, 2022, doi:10.3390/bs12090337_

Round 1
Reviewer 1 Report
- I believe that the theoretical framework should not only review the psychological interrelationship between residents and tourists but also contextualise the relevance of inhabitants as brand ambassadors in territory perception processes. There are important gaps in the theoretical review of city branding. Authors such as Florek, Anholt or Govers. In other words, the conceptual framework (figure 1) should be a square rather than a triangle to provide a global context and not only a specific one. Among the references in this field, only Kavaratzis (41) is to be found.
- A sample of 194 responses is too few to consider the results rigorous. Reliability is relative, which threatens the rigour of the research.
- The results are well presented, logical and comprehensible.
- In my - obviously subjective - view, the conclusion should be more comprehensive. The conclusion should contextualise the work and show the findings with which it contributes to knowledge, so that the reader, who has to read numerous papers on a daily basis, can get an idea of the mission, scope and content of the research in hand. These issues are partially synthesised in the discussion. However, in this block, more discussion with preceding research would be welcome.
Author Response
Dear reviewer,
Thank you for your detailed and constructive feedback. We have provided detailed responses to the comments below and included the changes that are reflected in the manuscript in red as well. Please see the attachment. We hope that this revision meets your expectations. Looking forward to your feedback in due course!

Reviewer 2 Report
The article fits the journal's scope. The topic of the article is interesting and relevant to the field. The style of writing is clear and concise. The results are significant and may be interesting for the readership of the journal. But, still the article needs some significant improvements in order to improve its quality and to be published in this journal. With some major revision, it may meet the standards of the journal.
- The Abstract of the paper should start with description of the topic and the the aim of the paper. After presentation of the study and the main results, it should point out importance of the study from different points of view, and state different stakeholders for whom the results of the study may be useful.
- Introduction section should be significantly reformulated- this section should actually introduce the topic itself and its importance from different points of view (it should exclude hypotheses etc. ).
- The structure of the paper should be improved.
- After Introduction new section Literature overview should be introduced. It should include more recent relevant publications (in the last five years). Special attention should be paid to presentation of other studies in this field and their results.
- In materials and methods new section should be introduced- Methodology of research and research questions and in this part it should present hypotheses etc.). Conceptual framework should be explained.
- In section Results- all performed analyses should be presented in a few sentences in order to understand their importance for the study and interpretation of the results.
- Although the results of the research are presented by tables and figures that are clear and easy to interpret and understand, l the results should be briefly explained after every table, figure etc., with a special focus on the proving of the hypotheses.
- The Discussion section should be significantly reformulated. It should include discussion of the all relevant results of the study, the analysis of the results, comparation with results of other studies in this field etc.
- The Conclusion section should be reformulated. It should summarise the main conclusions of the study and their importance for different stakeholders- from different points of view, as well as limitations of the research itself.
- The list of references should be improved with more up to date references (in the last five years).
Author Response

(The authors gave the same response as above.)

Reviewer 3 Report
Dear authors,
Thank you for sharing your findings. I find your manuscript “Exploring how the psychological resilience of residents of tour-2 ism destinations affected brand ambassador behavior during the COVID-19 pandemic” interesting and well written. Both the methodological part and the results are solid. I would have only few suggestions to the authors:
- - The literature review could be more critical
- - It is suggested to emphasise the scientific contribution of your study.
- - More information on the survey could be added (i.e., how respondents were approached, how the sample was defined, etc.)
- - Stronger discussion of findings in the context of prior literature would be beneficiary.
Good luck!
Author Response

(The authors gave the same response as above.)

Round 2
Reviewer 1 Report
I believe that with the reviewers' suggestions and the authors' modifications, this scientific communication has been significantly improved. I hope that it will have a positive impact on the community and that its findings will stimulate further research.
Reviewer 2 Report
Dear authors, thanks for your answers.
In my opinion, this version of the paper is much better than the previous one, the quality of the paper is significantly improved and as such it may give an important contribution to literature in this field. Good luck!